# Structural insights into the mechanism of overcoming Erm-mediated resistance by macrolides acting together with hygromycin-A

Chih-Wei Chen[1], Nadja Leimer [2], Egor A. Syroegin[1], Clémence Dunand[3], Zackery P. Bulman [4], Kim Lewis[2], Yury S. Polikanov [1,3,5] & Maxim S. Svetlov [3,5]

The ever-growing rise of antibiotic resistance among bacterial pathogens is one of the top healthcare threats today. Although combination antibiotic therapies represent a potential approach to more efficiently combat infections caused by susceptible and drug-resistant bacteria, only a few known drug pairs exhibit synergy/cooperativity in killing bacteria. Here, we discover that well-known ribosomal antibiotics, hygromycin A (HygA) and macrolides, which target peptidyl transferase center and peptide exit tunnel, respectively, can act cooperatively against susceptible and drug-resistant bacteria. Remarkably, HygA slows down macrolide dissociation from the ribosome by 60-fold and enhances the otherwise weak antimicrobial activity of the newest-generation macrolide drugs known as ketolides against macrolide-resistant bacteria. By determining a set of high-resolution X-ray crystal structures of drug-sensitive wild-type and macrolide-resistant Erm-methylated 70S ribosomes in complex with three HygA-macrolide pairs, we provide a structural rationale for the binding cooperativity of these drugs and also uncover the molecular mechanism of overcoming Erm-type resistance by macrolides acting together with hygromycin A. Altogether our structural, biochemical, and micro-biological findings lay the foundation for the subsequent development of synergistic antibiotic tandems with improved bactericidal properties against drug-resistant pathogens, including those expressing *erm* genes.

Although treatment with antibiotics remains an essential method to combat bacterial infections, the meteoric rise of antibiotic resistance among pathogenic microorganisms has severely restricted the utility of the existing arsenal of available drugs, creating a significant and ever-growing threat to healthcare[1]. Currently, the primary efforts aimed at the development of new antibiotics are either focused on the search for new natural products that possess antibacterial activity[2–5] or chemical derivatization of already known drugs to improve their antimicrobial efficacy[6–9]. Unfortunately, only a few of the newly dis-covered lead compounds will successfully pass preclinical and clinical

[1]Department of Biological Sciences, University of Illinois at Chicago, Chicago, IL 60607, USA. [2]Department of Biology, Northeastern University, Boston, MA 02115, USA. [3]Center for Biomolecular Sciences, University of Illinois at Chicago, Chicago, IL 60607, USA. [4]Department of Pharmacy Practice, University of Illinois at Chicago, Chicago, IL 60612, USA. [5]Department of Pharmaceutical Sciences, University of Illinois at Chicago, Chicago, IL 60607, USA. ✉e-mail: yuryp@uic.edu; msvet2@uic.edu

trials before finally being approved for clinical use[10]. Combinatorial antibiotic therapies, which utilize two or more drugs already approved for clinical use, offer a compelling alternative to improve and/or restore the efficacies of individual antibiotics used to treat infectious diseases caused by dangerous bacterial pathogens[11–15]. However, a recent systematic study of pairs of different antibiotics belonging to various classes unexpectedly showed that most of the tested drug combinations resulted in antagonism rather than cooperativity or synergy in early killing and long-term bacterial clearance[16]. Thus, knowledge-based approaches are needed to identify drug combinations whose binding and functional cooperativity translate into increased efficacy in bacterial growth inhibition and killing.

Many of currently used antibiotics inhibit protein synthesis in pathogenic microorganisms by targeting their 70S ribosomes, which are complex macromolecular machines responsible for this process[17,18]. The catalytic peptidyl transferase center (PTC) located at the heart of the large ribosomal subunit (50S in bacteria) polymerizes amino acids into polypeptides in the order specified by the nucleotide sequences of the translated mRNAs. Newly-made proteins emerge from the ribosome through the nascent peptide exit tunnel (NPET) that spans the body of the large ribosomal subunit beginning at the PTC and ending on the opposite side of the large ribosomal subunit facing the cytoplasm. Many classes of ribosome-targeting antibiotics bind in the PTC or NPET and interfere with the normal ribosome functioning and production of the bacterial proteome[17,18]. Macrolides represent one of the largest and clinically significant groups of antibiotics that bind the exit tunnel of the bacterial ribosome in a specific site formed by nucleotides A2058 and A2059 of the 23S rRNA (*E. coli* numbering used throughout the text) and allosterically inhibit polymerization of specific amino acid motifs by adjacent PTC resulting in selective arrest of synthesis of a subset of cellular proteins and stop bacterial growth[19].

Among bacterial pathogens, one of the most frequent and clinically relevant mechanisms of resistance to macrolides is based on mono- or dimethylation of the N6 position of adenine 2058 residue (A2058) of the 23S rRNA by Erm methyltransferases[20–23]. In addition to macrolides, the same modification confers resistance to other structurally unrelated classes of ribosome-targeting antibiotics, such as lincosamides and streptogramin B (hence this type of resistance is known as MLS$_B$), that also bind in the NPET[23,24]. Significant efforts have been made to create macrolides capable of overcoming this type of bacterial drug resistance, culminating in the development of semi-synthetic ketolides, such as telithromycin (TEL) and solithromycin (SOL)[25]. Unlike their natural predecessor erythromycin (ERY), the semi-synthetic ketolides carry a keto-group instead of a cladinose sugar and an extended alkyl-aryl side chain that stacks upon A752-U2609 base-pair of the 23S rRNA and contributes to the increased affinity and slow dissociation from the bacterial ribosome. The increased residence time of ketolides on the bacterial ribosome results in their high cidality against a variety of drug-susceptible Gram-positive pathogens[26,27]. Importantly, unlike the parent compound, TEL and SOL manifest residual binding to the Erm-methylated ribosomes and inhibit growth of *erm*-positive macrolide-resistant bacterial strains[28]. However, the initially bactericidal ketolides become only bacteriostatic against *erm*-positive pathogens inducing their dormancy[26].

Another approach to improve the activity of macrolides against susceptible and resistant bacteria, including Erm-expressing strains, would be to utilize them in combination with other drugs that bind to an adjacent but non-overlapping site in the 70S ribosome (such as PTC). Binding of the second drug could stimulate the binding of a macrolide either directly (through interactions with the ribosome-bound macrolide molecule) or allosterically (through induction of conformational re-arrangements in the ribosome). There are several well-known pairs of ribosome-targeting antibiotics that simultaneously bind to the bacterial ribosome and interfere with protein synthesis. For example, streptogramins A and B, such as dalfopristin and quinupristin, bind to adjacent sites in the PTC and NPET, respectively, and synergistically inhibit peptide elongation[29,30]. These molecules display enhanced affinity for the ribosome when in combination[31]. Moreover, while the individual streptogramins are bacteriostatic, they become bactericidal if present together[32]. Another synergistic pair of ribosome-targeting antibiotics are lankacidin and lankamycin, which also bind simultaneously to the neighboring sites in the PTC and NPET, respectively, and exert their synergistic inhibition of protein synthesis[33]. Thus, other ribosome-targeting antibiotics that (1) bind adjacent to macrolides, such as PTC inhibitors, and (2) can co-exist on the ribosome together with macrolides potentially might enhance protein synthesis inhibition and bacterial killing caused by macrolides.

In this work, we set to determine whether any of the PTC-binding drugs can act cooperatively (or even synergistically) with macrolides. First, in silico structural analysis predicted that out of several PTC inhibitors, whose structures in complex with the 70S ribosome are available, only hygromycin A (HygA) could co-exist with the macrolides and ketolides. Then using biochemical techniques, we show that, unlike other tested PTC-targeting antibiotics competing with macrolides, only HygA cooperatively binds ribosomes with NPET-targeting macrolides and slows down their dissociation. Moreover, HygA potentiates macrolide's efficacy in bacterial growth inhibition and early killing. Most excitingly, we found cooperative action of HygA and ketolides against a macrolide-resistant strain of *Streptococcus pneumoniae* constitutively expressing an *erm* methyltransferase gene. Finally, by determining X-ray crystal structures of both unmodified wild-type and Erm-methylated 70S ribosomes in complex with one of the three HygA-macrolide pairs, we provide a rational explanation for the binding cooperativity of these drugs. Altogether our results suggest that HygA enhances the antibacterial activity of macrolides. Our study provides pivotal information for the rational structure-based design of tandems of the PTC inhibitors acting together with macrolides that possess even higher activity against drug-resistant pathogens.

## Results

### Hygromycin A binds 70S ribosome simultaneously with macrolides

We reasoned that PTC-targeting antibiotics could potentially be cooperative (or even synergistic) with macrolides only if their binding sites on the ribosome are not overlapping and, thus, they can bind to the ribosome simultaneously. Therefore, to identify candidates, first, we checked if any of the PTC-binding inhibitors could, in principle, co-exist with macrolides on the bacterial ribosome. To this end, we performed in silico analysis of the available structures of 70S-bound PTC-targeting drugs−hygromycin A (HygA)[34], A201A[34], clindamycin[35], and linezolid[36]−for their possible steric overlaps with the ribosome-bound macrolide ERY (Fig. 1a–d). Our previous structural work revealed a direct steric clash between a classic PTC inhibitor chloramphenicol (CHL) and a macrolide antibiotic ERY, rationalizing their mutually-exclusive competitive binding to the 70S ribosome[37]. Thus, we used CHL as a negative control. Superposition of the structures showed that among all analyzed PTC inhibitors, only HygA does not overlap with ERY (Fig. 1a), suggesting that the two drugs may be able to bind simultaneously to the ribosome. To test this prediction, we measured the binding of radiolabeled [$^{14}$C]-ERY to the 70S ribosomes in the presence of increasing concentrations of analyzed PTC-acting drugs. In agreement with our in silico structure-based predictions, the competition binding assay revealed that out of all tested PTC inhibitors, only HygA does not displace ERY from the 70S ribosomes (Fig. 1e, yellow graph). Interestingly, at high concentrations, HygA even stimulates the binding of [$^{14}$C]-ERY to the 70S ribosomes pointing to the possible binding cooperativity between these drugs.

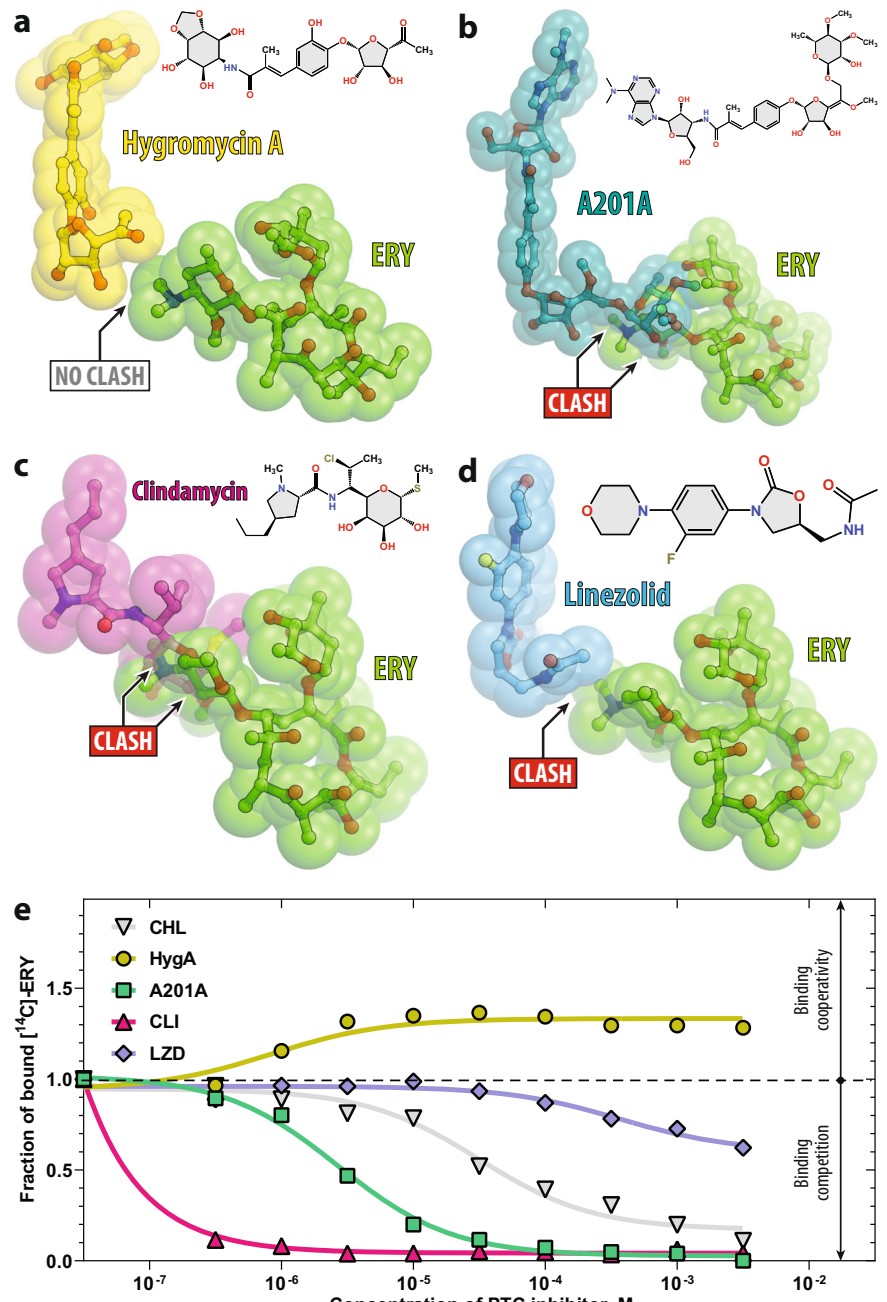

**Fig. 1 | Comparison of the macrolide binding site with those of various PTC-targeting antibiotics.** Superposition of the structures of the ribosome-bound ERY (red, PDB entry 6XHX[20] [https://doi.org/10.2210/pdb6XHX/pdb]) and **a** hygromycin A (yellow, PDB entry 5DOY[34] [https://doi.org/10.2210/pdb5DOY/pdb]), **b** nucleoside antibiotic A201A (teal, PDB entry 4Z3S[34] [https://doi.org/10.2210/pdb4Z3S/pdb]), **c** clindamycin (magenta, PDB entry 4V7V[35] [https://doi.org/10.2210/pdb4V7V/pdb]), or **d** linezolid (light blue, PDB entry 7S1G[65] [https://doi.org/10.2210/pdb7S1G/pdb]). All structures of ribosome-bound antibiotics were aligned based on domain V of the 23S rRNA. Note that only hygromycin A does not sterically clash with ribosome-bound ERY, while A201A, clindamycin, and linezolid overlap with the desosamine sugar of ERY. **e** Competition binding assay to assess the release of

[14C]-radiolabeled ERY from the 70S ribosomes in the presence of increasing concentrations of one of the PTC-targeting antibiotics shown in (**a**)−(**d**). Chloramphenicol (CHL, gray) is used as a positive control known to compete with ERY[37]. The amount of [14C]-ERY associated with the ribosomes in the absence of a competitor drug is arbitrarily assigned as 1.0 (dashed line). Under experimental conditions, this corresponds to ~50% of the 70S ribosomes bound to [14C]-ERY. The measurements were repeated twice with similar results. Source data are provided as a Source Data file. Note that, at high concentrations, only HygA (yellow) stimulates additional binding of ERY to the 70S ribosome, whereas A201A (teal), clindamycin (magenta), linezolid (light blue), or chloramphenicol (gray) cause its dissociation.

## Hygromycin A slows down dissociation of macrolides from the ribosome

Next, we checked whether HygA and ERY exhibit binding synergy by measuring the affinity of radiolabeled [14C]-ERY to the 70S ribosome in the absence and presence of unlabeled HygA (Fig. 2a). Unexpectedly, direct measurements of the equilibrium dissociation constants show

that the affinity of ERY to the 70S ribosomes increases only ~1.6-fold in the presence of HygA, which is not a statistically significant change ($K_{d(ERY)} = 10.21 \pm 5.95$ nM and $K_{d(ERY + HygA)} = 5.91 \pm 1.65$ nM). However, since the equilibrium dissociation constants reflect the ratios of the on- and off-rates of ligand binding, the observed comparable affinities could mask the significantly different kinetics of drug association with

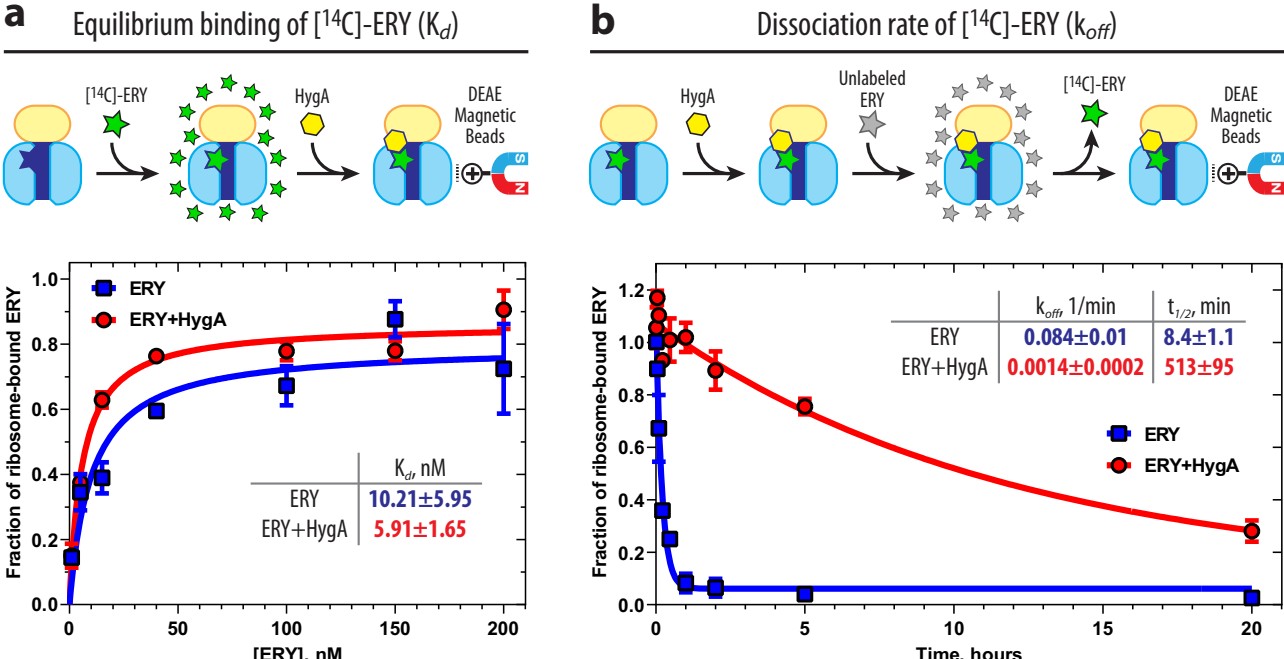

**Fig. 2 | Effects of hygromycin A on binding properties of erythromycin to the 70S ribosome. a** Equilibrium binding of radiolabeled [$^{14}$C]-ERY to determine dissociation constants ($K_d$). *S. pneumoniae* 70S ribosomes pre-equilibrated with increasing concentrations of [$^{14}$C]-ERY in the absence (blue curve) and presence (red curve) of 100 μM HygA were isolated using positively-charged DEAE magnetic beads, and the amount of remaining ribosome-associated radioactivity was measured in a scintillation counter. Error bars show mean standard deviation of three independent measurements. Note that HygA does not significantly improve the binding affinity of ERY to the ribosome. **b** Kinetics of [$^{14}$C]-ERY dissociation from *S. pneumoniae* 70S ribosomes saturated with [$^{14}$C]-ERY in the absence (blue curve)

and presence (red curve) of 100 μM HygA. The amount of remaining ribosome-associated radioactivity was measured at different time points after diluting ribosomes with molar excess of non-radiolabeled ERY. Experimental data were fitted with a one-phase exponential function that yielded dissociation rate constants ($k_{off}$) of 0.084 ± 0.01 1/min for ERY alone and 0.0014 ± 0.0002 1/min for ERY in the presence of HygA. The constants were used to calculate the half-lives ($t_{1/2}$) of the complexes (shown in the figure). Error bars show mean standard deviation of two independent measurements. Source data are provided as a Source Data file. Note that HygA significantly (60-fold) slows down the dissociation rate of ERY from the 70S ribosome.

and dissociation from the target. Therefore, using the previously published technique[26], we measured the dissociation rates of radiolabeled [$^{14}$C]-ERY from the 70S ribosomes in the absence and presence of HygA (Fig. 2b). To this end, the amount of radioactivity on the 70S ribosomes was quantified at different time points after adding an excess of unlabeled ERY. In agreement with published data[26], dissociation of ERY from the ribosomes occurs relatively fast, with a half-life of ~8 min (50% of ribosome-bound ERY dissociates in ~8 min). Remarkably, the addition of HygA dramatically slows down ERY dissociation increasing its half-life on the ribosome by 60-fold to 9 h. These data demonstrate the ability of PTC inhibitor HygA and NPET-acting macrolide ERY to bind bacterial ribosomes cooperatively. Moreover, using a toe-printing assay to map the position of the ribosomes along mRNA template, we found that cooperative binding of HygA and ERY leads to translational arrest at the start codon (Supplementary Fig. 1). Therefore, in contrast to the context-specific action of ERY during the elongation phase of translation[38], the HygA-ERY pair effectively, and likely indiscriminately, inhibits protein synthesis at the initiation stage.

## Hygromycin A structurally predisposes ribosomes to macrolide binding

To gain further insight into the molecular basis of HygA-macrolide cooperativity, we set out to determine a series of high-resolution X-ray crystal structures of 70S ribosome in complex with HygA-macrolide pairs. To this end, we co-crystallized wild-type (WT) 70S ribosomes from the Gram-negative bacterium *Thermus thermophilus* (*Tth*) with HygA and one of the three macrolides (ERY, azithromycin (AZI), or TEL) and determined their structures at 2.5–2.6 Å resolution (Fig. 3 and

Supplementary Table 1). In this experiment, we used *Tth* 70S ribosomes complexed with *E. coli* protein Y (PY) as a tool to obtain structures of higher resolution[20,39–42]. Since the binding site of PY is located on the small ribosomal subunit, where it overlaps with mRNA and all three tRNAs, it does not interfere with the binding of HygA in the PTC nor macrolides in the NPET of the large ribosomal subunit. As a result, by using ribosome-PY complexes, we were able to obtain higher resolution and overall better quality electron density maps of the ribosome-bound drugs than it was possible otherwise.

The obtained electron density maps revealed all three drug pairs bound to their previously known binding sites on the 70S ribosome (Fig. 3a–c). Consistent with the previous structure[34], HygA binds in the A-site cleft of the PTC on the large ribosomal subunit with its aromatic ring sandwiched between nucleotides A2451 and C2452 of the 23S rRNA (Fig. 4a–c), and is oriented such that the aminocyclitol group protrudes toward nucleotide C2573, while the fucofuranose ring extends into the ribosome exit tunnel (Fig. 4b). Most importantly, the presence of a neighboring macrolide (either ERY, AZI, or TEL) does not affect the binding position of HygA, in which it interferes with the full accommodation of the amino acid moiety of an incoming aminoacyl-tRNA (aa-tRNA) into the ribosomal A site, resulting in inhibition of peptide bond formation[34]. Interestingly, in the presence of HygA, macrolides also appear in their canonical binding pocket within the NPET that is formed by nucleotides A2058 and A2059 of the 23S rRNA (Fig. 4d). In all three structures of macrolides in combination with HygA, the desosamine moiety of a macrolide forms two hydrogen bonds (H-bonds), which were suggested to be crucial for macrolide binding[20]: one between the hydroxyl group of the desosamine and the N1 atom of the nucleotide A2058, and the second one between the

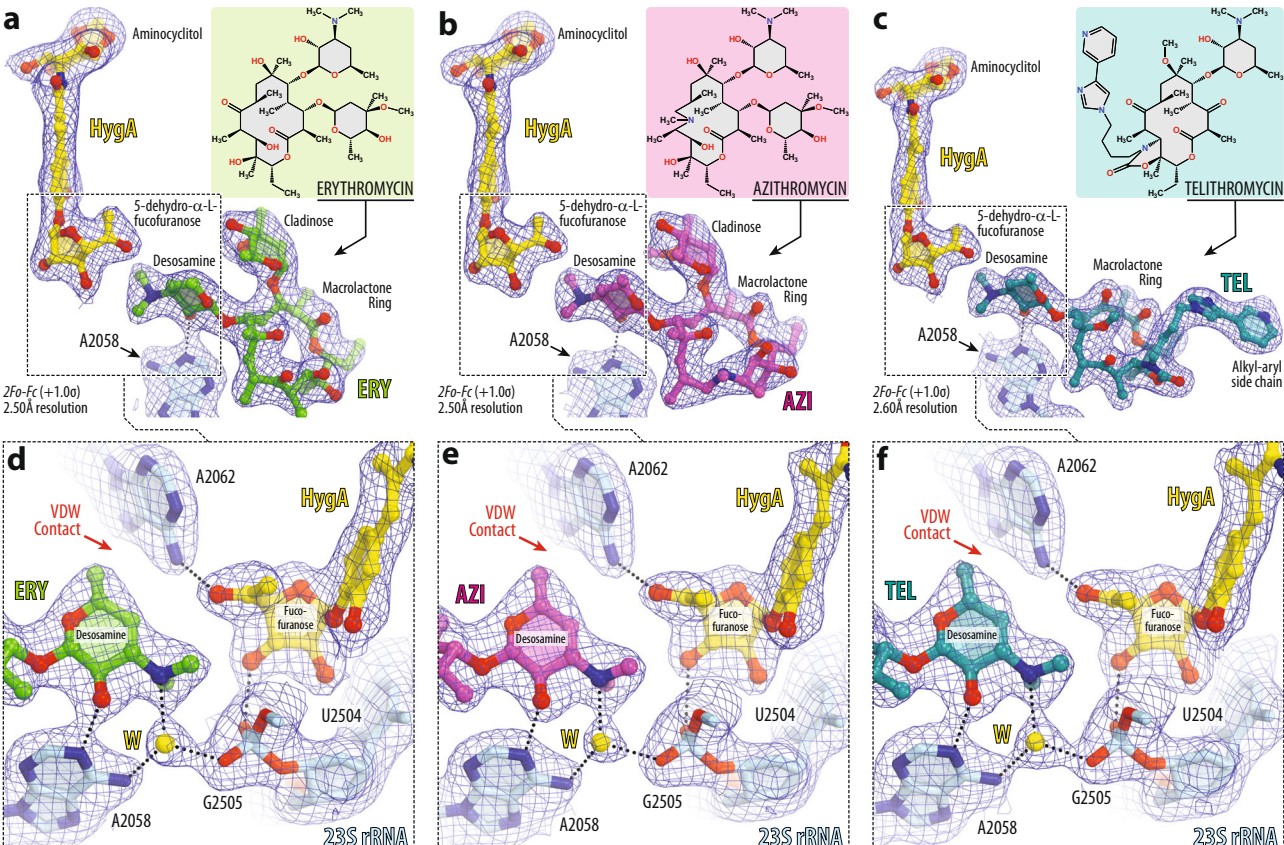

**Fig. 3 | Electron density maps of three macrolides bound to the *T. thermophilus* 70S ribosome together with hygromycin A.** $2F_o$-$F_c$ electron difference Fourier maps of hygromycin A (HygA, yellow) and either erythromycin (**a**, ERY, green), or azithromycin (**b**, AZI, magenta), or telithromycin (**c**, TEL, teal). The refined models of antibiotics are displayed in their respective electron density maps after the refinement (blue mesh). The overall resolution of the corresponding structures and the contour levels of the depicted electron density maps are shown in the bottom left corner of each panel. Chemical structures of corresponding macrolides are shown as insets. Close-up views of high-resolution electron density maps of ribosome-bound HygA together with ERY (**d**), AZI (**e**), or TEL (**f**) interacting with the nucleotides of the 23S rRNA (light blue). Note that the dimethylamino group of all macrolides is rotated toward nucleotide A2058 and forms an H-bond with a water molecule (W, yellow) tightly coordinated by the exocyclic N6-amino group of A2058 and the phosphate of G2505. Nitrogens are colored blue; oxygens are red (except for the water). Also, note that the atoms of desosamine group of a macrolide form van der Waals contacts with the fucofuranose moiety of HygA and nucleobase A2062 (red arrows).

dimethylamino group of desosamine sugar and the N6 atom of A2058 via a water molecule that is also coordinated by the phosphate of G2505 (Fig. 3d–f). In full agreement with our initial in silico prediction (Fig. 1a), our structures show that HygA can bind to the WT bacterial ribosome simultaneously with ERY, AZI, or TEL antibiotics.

In the case of synergistic streptogramins A and B, drug-induced re-arrangement of the nucleotide A2602 caused by one of the two anti-biotics can enhance binding of the second one via additional stacking and/or H-bond interactions[30,43–46]. Interestingly, in our structures of the 70S ribosome in complex with either of the HygA-macrolide pairs, nucleotide A2062 of the 23S rRNA is oriented exactly the same way as in the previous independent structures of 70S-HygA[34] and 70S-ERY[20] complexes (Supplementary Fig. 2). Compared to its position in the drug-free ribosome, nucleotide A2062 rotates by ~160 degrees and forms a Hoogsteen base-pair with the nucleotide m²A2503 in the presence of HygA, or a macrolide, or both. Ribosome-bound HygA favors this rotated conformation of A2062 via direct H-bond with the N6 atom of this nucleotide (Fig. 4b). In contrast, ERY does so by establishing a CH-π interaction between the methyl group of its desosamine sugar and the aromatic nucleobase of A2062 (Fig. 4d and Supplementary Fig. 3a). Importantly, in either case, nucleotide A2062 adopts precisely the same conformation. Therefore, it is tempting to attribute the observed HygA-ERY binding cooperativity (Fig. 1e) to the observed re-orientation of nucleotide A2062: regardless of which of the two drugs causes this

re-arrangement in the first place (HygA or macrolide), it implicitly organizes the second drug's binding pocket, making its binding to the ribosome more favorable, rationalizing the observed HygA-ERY binding cooperativity. Although the degree of A2062 rotation varies between the HygA-macrolide pairs and $S_A$/$S_B$ streptogramins, the general mechanism of binding cooperativity seems to be conceptually the same, in which nucleotide A2062 serves as a mediator between the A-site-bound and the NPET-bound antibiotics. Unlike streptogramins, which do not promote the formation of A2062–A2503 Hoogsteen base-pair, re-arrangement of A2062 induced by either HygA or ERY leads to the formation of such a base-pair (Fig. 4b, d).

Another factor contributing to the cooperativity between HygA and a macrolide is their possible direct interaction with each other on the ribosome. Although ribosome-bound HygA and either of the macrolides (ERY, AZI, or TEL) do not form H-bonds with each other, their fucofuranose and desosamine groups, respectively, come into physical proximity and form van der Waals contacts (Supplementary Fig. 3b). This relatively weak interaction between the two drugs on the ribosome is unlikely to improve their affinity significantly, in agreement with our biochemical data (Fig. 2a). However, ribosome-bound HygA can prevent macrolide dissociation by sterically occluding the access path to its binding pocket from the PTC side, rationalizing the increased dwell time of ERY on the ribosome in the presence of HygA (Fig. 2b).

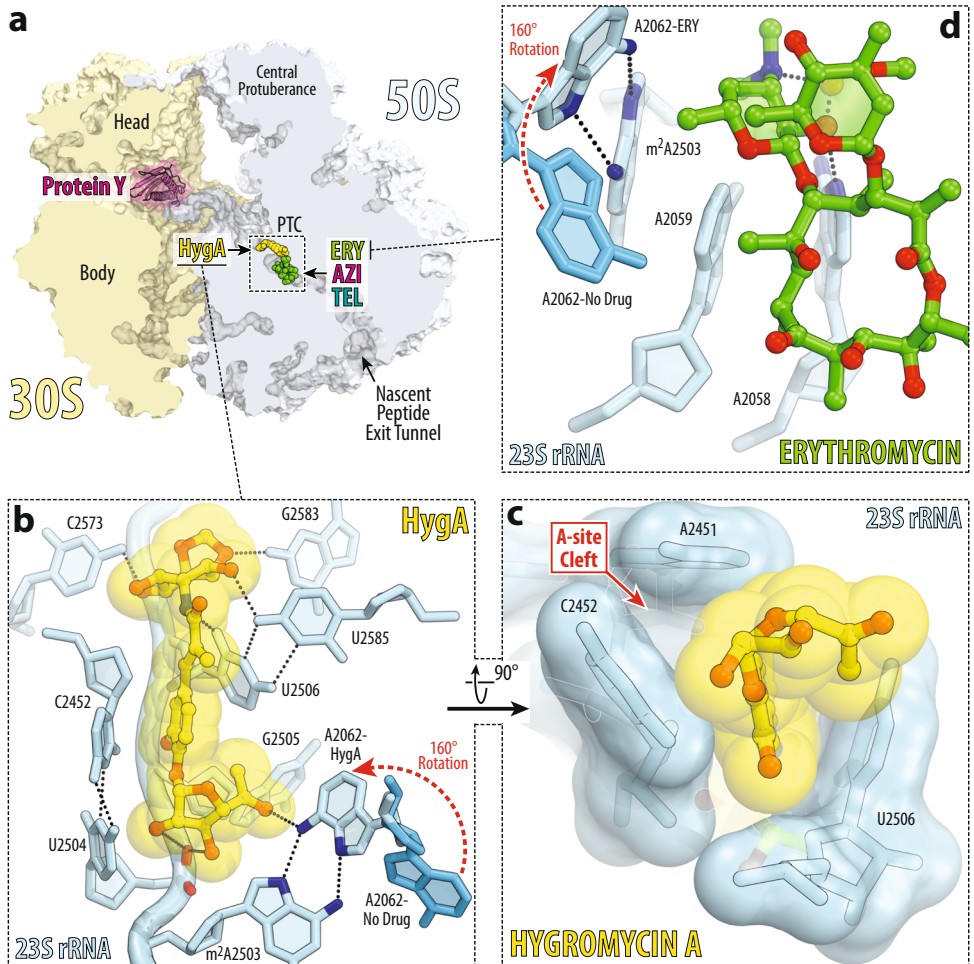

**Fig. 4 | Structure of HygA and ERY simultaneously bound to the 70S ribosome.**
**a** Overview of the *T. thermophilus* 70S ribosome structure with bound HygA (yellow) and ERY (green) viewed as a cross-cut section through the nascent peptide exit tunnel (NPET). The 30S subunit is shown in light yellow; the 50S subunit is in light blue; ribosome-bound protein Y is colored magenta. The binding positions of AZI and TEL are nearly the same as ERY. Close-up views of the HygA (**b**, **c**) and ERY (**d**) binding sites in the PTC and NPET of the 70S ribosome, respectively (*E. coli* numbering of the rRNA nucleotides is used throughout). H-bond interactions are indicated with dotted lines. Note that by forming an H-bond with the base of nucleotide A2062 of the 23S rRNA (light blue), HygA causes rotation of this nucleotide by ~160 degrees to form Hoogsteen base-pair with the m²A2503 residue of the 23S rRNA (red dashed arrow). The binding of ERY, as well as other macrolides, causes the same characteristic rotation of nucleotide A2062. The unrotated conformation of A2062 observed in the absence of either drug is shown in blue (PDB entry 4Y4O[40] [https://doi.org/10.2210/pdb4Y4O/pdb]).

## Hygromycin A potentiates macrolides' ability to kill susceptible and resistant bacteria

The observed cooperative binding of HygA and macrolides to the 70S ribosome suggests that HygA-macrolide combinations could potentially possess higher antibacterial potency than their monotherapies. To check whether the cooperative binding of HygA with macrolides translates into their enhanced antibacterial action, we tested the ability of these drugs alone and in pairs to inhibit bacterial growth. To this end, we used a standard checkerboard assay[47] that represents a two-dimensional array of bacterial growth tests in which increasing concentrations of one drug are checked against the increasing concentrations of the other. The range of tested concentrations for each drug typically goes from zero to the minimal inhibitory concentration (MIC) for a particular antibiotic (Supplementary Table 2). The result of this assay for a given pair of drugs acting against a specific bacterial strain is expressed as a fractional inhibitory concentration (FIC) index, where an FIC index value of ≤0.5 reflects the antimicrobial synergy between two drugs, whereas FIC index values between 0.5 and 1.0 is defined as additivity[48]. Conversely, FIC index values >1.0 indicate antagonism in drug action. For the checkerboard assay, we used clinically relevant strains of bacteria sensitive to both HygA and macrolides, such as the Gram-positive *Streptococcus pneumoniae* Cp2000 strain. For this strain, the HygA-ERY pair had a minimal FIC index of 0.63 (Supplementary Fig. 4a), which can be considered as cooperative action. Interestingly, if combined with macrolides of second and third generations, such as AZI, TEL, or SOL, HygA inhibited the growth of *S. pneumoniae* even more effectively with FIC index values of 0.5–0.6 (Supplementary Fig. 4a). These FIC index values were at the borderline of additive vs. synergistic action. Recently, HygA was shown to be especially active and selective against *Borrelia burgdorferi*, a Gram-negative bacterium from the spirochete class that causes Lyme disease[49]. Since *B. burgdorferi* is also sensitive to macrolides (Supplementary Table 2), we tested the activity of the HygA-ERY pair against this species and found a minimal FIC index of 0.5, again suggesting solid antimicrobial cooperativity (but not quite synergy) of these drugs (Supplementary Fig. 4b).

Previously, it was shown that the slow kinetics of dissociation of some macrolides, such as ketolide TEL, from the bacterial ribosome strongly correlates with their ability to kill drug-susceptible bacteria (bactericidal effect) rather than stop their growth (bacteriostatic effect)[26]. Since HygA significantly increases the dwell time of ERY on the ribosome (Fig. 2b), we wondered if HygA can potentiate macrolide's

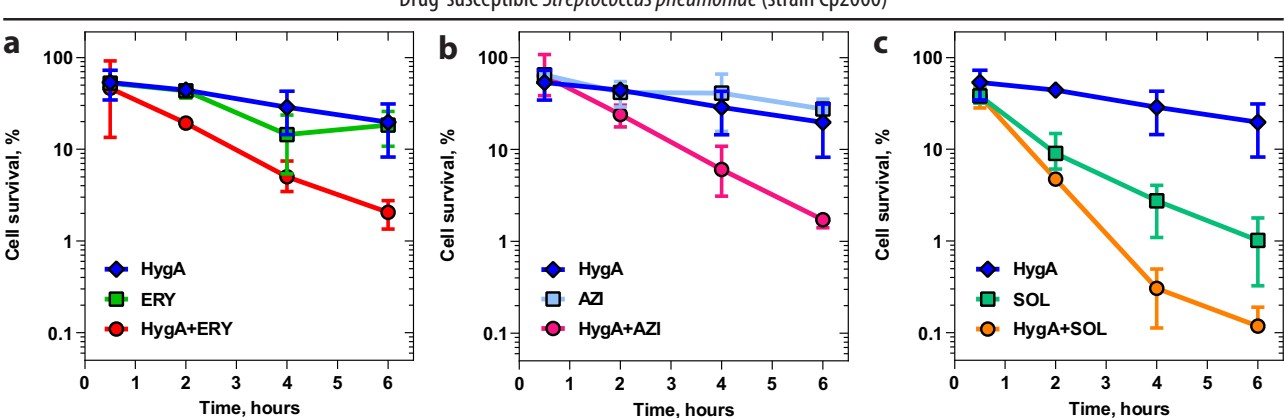

**Fig. 5 | Hygromycin A potentiates bactericidal properties of macrolide antibiotics.** Time-kill assays using drug-susceptible Cp2000 strain of *S. pneumoniae* exposed during various times to antibiotic concentrations at 4x MIC of hygromycin A (HygA, blue plot, 20 μg/ml), macrolides erythromycin (**a**, ERY, green plot, 0.25 μg/ml), azithromycin (**b**, AZI, light blue plot, 0.5 μg/ml), or ketolide solithromycin (**c**, SOL, teal plot, 0.04 μg/ml) alone and in combination with HygA (red, magenta, and orange plots, respectively). The initial number of viable cells (colony-forming units, CFUs) before the addition of drug(s) was arbitrarily assigned to 100%. Error bars show mean standard deviation of 2–4 independent measurements. Source data are provided as a Source Data file.

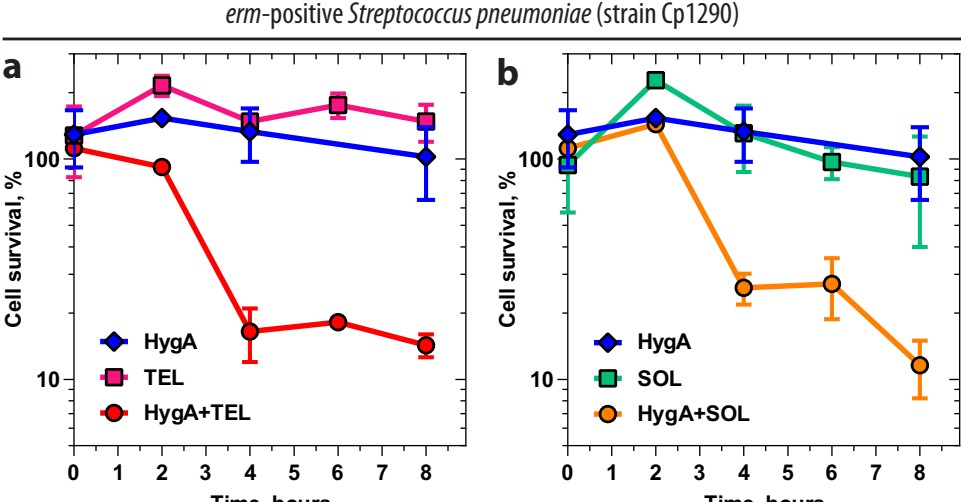

**Fig. 6 | Hygromycin A restores the bactericidal activity of ketolides against Erm-expressing bacteria.** Time-kill assays using *erm*-positive macrolide-resistant Cp1290 strain of *S. pneumoniae* exposed during various times to antibiotic concentrations at 4x MIC of hygromycin A (HygA, blue plot, 20 μg/ml), ketolides telithromycin (**a**, TEL, magenta plot, 2 μg/ml) or solithromycin (**b**, SOL, teal plot, 4 μg/ml) alone and in combination with HygA (red and orange plots, respectively). The initial number of viable cells (colony-forming units, CFUs) before the addition of drug(s) was arbitrarily assigned to 100%. Error bars show mean standard deviation of 2–5 independent measurements. Source data are provided as a Source Data file.

cidality. Using standard time-kill assays, we found that either HygA or ERY are bacteriostatic against the drug-susceptible *S. pneumoniae* strain Cp2000, allowing many cells to survive even after 6-h exposure to concentrations 4x MIC of each drug (Fig. 5a, blue and green plots). However, unlike individual drugs, the HygA-ERY pair kills a significant fraction (>1 log reduction) of the cell population (Fig. 5a, red plot). A similar result is observed with the HygA-AZI pair (Fig. 5b, orange plot). Thus, combination with HygA renders otherwise bacteriostatic macrolides ERY and AZI with the ability to kill bacterial cells. Moreover, HygA further improves the killing activity of an already bactericidal ketolide, SOL, against *S. pneumoniae* Cp2000 (Fig. 5c, teal vs. orange plots)[26]. Importantly, tetracycline—another bacteriostatic ribosome-targeting antibiotic that binds to a distant site on the small ribosomal subunit— is unable to enhance bacterial growth inhibition and killing properties of macrolides or ketolides (Supplementary Fig. 5), suggesting that the observed potentiation of bacterial killing stems from the direct interactions of HygA with a macrolide on the ribosome.

Expression of Erm methyltransferases that N6-dimethylate nucleotide A2058 of the 23S rRNA results in strong resistance to macrolides in bacteria[20–23]. Therefore, we were curious if HygA could potentiate macrolides' otherwise weak antimicrobial activity against Erm-expressing strains. In a checkerboard assay, HygA-TEL and HygA-SOL combinations exhibited additivity against Erm-expressing *S. pneumoniae* cells with FIC index values in the range of 0.6–0.8 (Supplementary Fig. 4c). Comparison of previously published MICs between macrolide-sensitive Cp2000 and macrolide-resistant ErmA-expressing Cp1290 strains of *S. pneumoniae* showed that A2058-N6-dimethylation has a 16,000-fold effect for ERY and a more modest (but still significant) -100-fold effect for the newest-generation macrolide antibiotics—ketolides (such as TEL or SOL)[26]. Unlike ERY and AZI, which are completely inactive against macrolide-resistant *S. pneumoniae* Cp1290 cells, ketolides TEL and SOL can inhibit their growth (Fig. 6, magenta and teal plots, respectively), which is, however, bacteriostatic and not bactericidal as in the case with WT cells (Fig. 5)[26]. Remarkably,

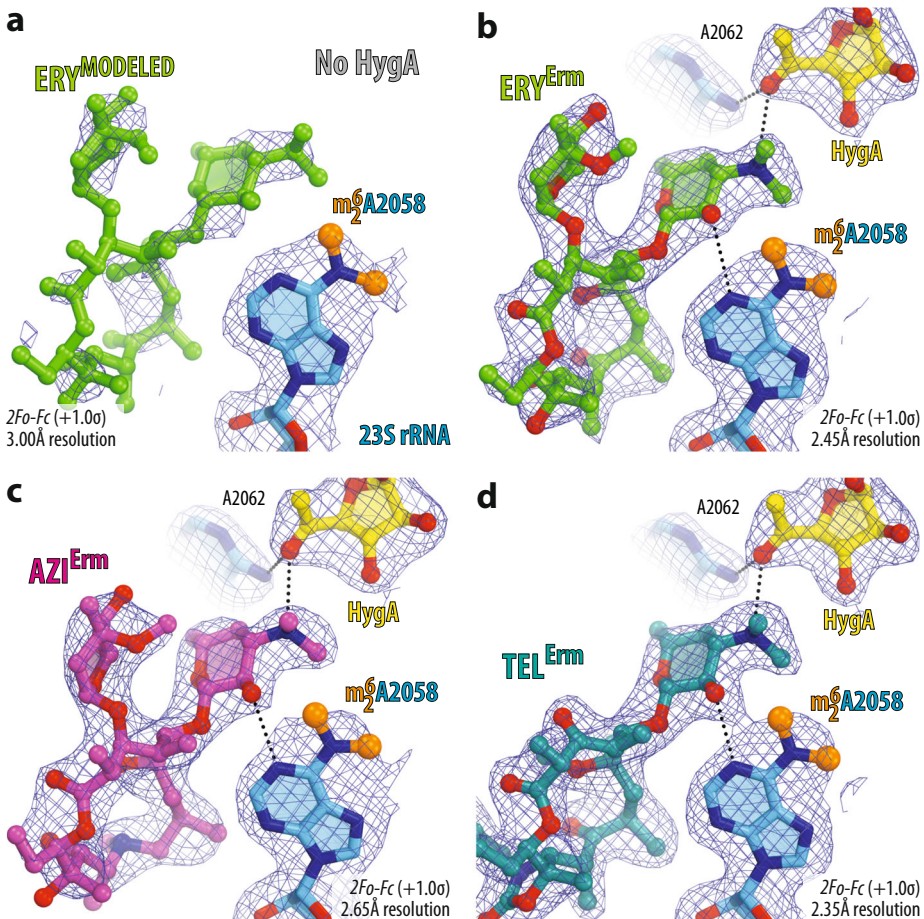

**Fig. 7 | Electron density maps of three macrolides bound to the Erm-modified 70S ribosome in the presence of hygromycin A.** Electron density maps of macrolides erythromycin (**a**, **b**), azithromycin (**c**), or telithromycin (**d**) in the absence (**a**) and presence of HygA (**b**–**d**) bound to the Erm-modified *Tth* 70S ribosome containing N6-dimethylated residue A2058 (blue with methyl groups highlighted in orange) in the 23S rRNA. H-bonds are depicted with dotted lines.

in our time-kill assays, HygA partially restores the killing activity of TEL and SOL against *erm*-positive *S. pneumoniae* Cp1290 cells, showing a nearly 10-fold decrease in the observed number of colony-forming units (CFUs) after 8-h exposure to the HygA-ketolide pairs at 4x MICs for both drugs in each tested combination (Fig. 6, red and orange plots, respectively). While our results demonstrate that HygA potentiates the killing activity of macrolides against drug-resistant bacteria, it is unclear whether or not these effects are due to canonical macrolide binding to the Erm-methylated 70S ribosome potentiated by HygA.

### Hygromycin A enables macrolide binding to the Erm-methylated ribosome

The activity of HygA-macrolide pairs against bacteria harboring *erm* resistance genes prompted us to determine X-ray crystal structures of HygA-macrolide combinations bound to the Erm-methylated 70S ribosome containing N6-dimethylated residue $m_2^6A2058$. Using our recent methodology, Erm-modified 70S ribosomes were isolated from *Tth* cells expressing the *erm*-like gene from *Bifidobacterium thermophilum*[20]. Co-crystallization of these Erm-methylated ribosomes with PY and the same HygA-macrolide pairs as before (HygA-ERY, HygA-AZI, and HygA-TEL) provided crystals diffracting to 2.35–2.65 Å resolution (Fig. 7 and Supplementary Table 1). Remarkably, the observed electron-density maps revealed strong peaks corresponding to HygA and the macrolides (ERY, AZI, or TEL) bound to the A2058-N6-dimethylated ribosomes (Fig. 7b–d, respectively), allowing us to build molecular models unambiguously. As a control experiment, we tried to

obtain the same structures in the absence of HygA and observed only fragmented and weak electron density peaks corresponding to the ribosome-bound ERY (Fig. 7a and Supplementary Fig. 6a). In the case of TEL alone (without HygA), the observed electron density is somewhat stronger and less fragmented than that of ERY (Supplementary Fig. 6c), in agreement with the residual binding affinity of ketolides for the Erm-methylated ribosome. We shall note, however, that the extent of A2058-N6-dimethylation in Erm-modified 70S ribosomes used for crystallization reached ~60%[20]. Therefore, the observed residual electron density of ERY and TEL on the Erm-methylated ribosome (Fig. 7a and Supplementary Fig. 6a, c) likely reflects binding of macrolides to the mono- and/or unmethylated 70S ribosomes present in the sample.

In our recent study, we discovered a cryptic water molecule that appears to be playing a pivotal role in binding of macrolides to the ribosome[20]. In the WT ribosome, this water molecule is coordinated by the nucleobase of A2058 and the phosphate of G2505 and serves as a binding partner for the dimethylamino moiety of the functionally critical desosamine sugar of a macrolide (Fig. 3d–f). In the Erm-methylated ribosome, this water molecule is displaced due to direct sterical hindrance with the methyl groups appended to the A2058 residue, rationalizing the significantly reduced affinity of macrolides to the Erm-modified ribosome. However, the overall binding positions of ERY, AZI, or TEL on the A2058-methylated 70S ribosome in the presence of HygA appear to be nearly identical to their binding sites in the WT ribosome (Fig. 8a, b and Supplementary Fig. 7). How is it possible, given that the key water molecule required for macrolide binding must be absent from the Erm-methylated ribosome? Interestingly, the

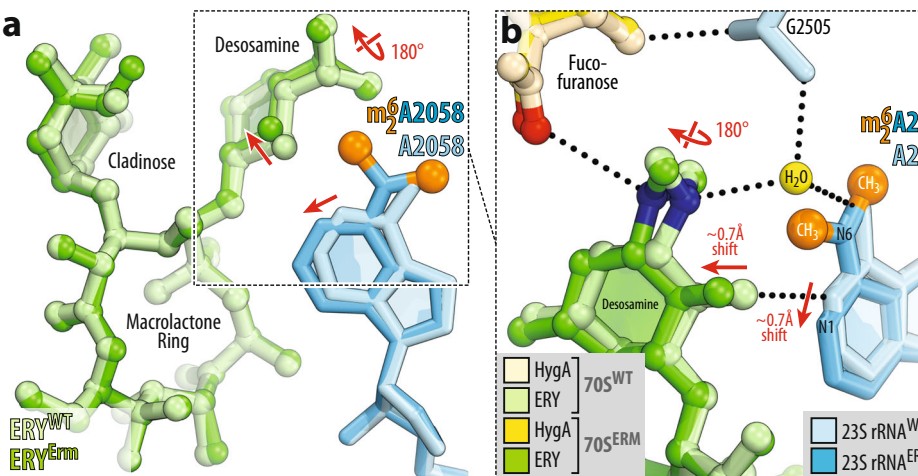

**Fig. 8 | Comparison of structures of erythromycin bound to the WT and Erm-modified 70S ribosome in the presence of hygromycin A. a, b** Superposition of ERY (light green) in complex with the WT 70S ribosome containing unmodified residue A2058 (light blue) and the structure of ERY (green) in complex with the Erm-modified 70S ribosome containing N6-dimethylated residue A2058 (blue with methyl groups highlighted in orange). Note that, while the overall position of ribosome-bound ERY is nearly identical in the two structures, N6-dimethylation of A2058 results in a 180-degree rotation of the dimethylamino group of desosamine away from A2058 nucleobase toward the fucofuranose moiety of HygA and formation of a direct H-bond with it. Also, note that N6-dimethylation of A2058 causes a small shift of ERY's desosamine moiety away from the nucleotide, potentially weakening the H-bond between the 2′-OH of macrolide and the N1 atom of A2058 (red arrows).

dimethylamino moiety of the desosamine sugars of ERY, AZI, or TEL rotates by ~180° around the C-N bond and finds a new H-bonding partner—the carbonyl oxygen of the fucofuranose ring of ribosome-bound HygA (Fig. 8a, b and Supplementary Fig. 7). Since the tertiary amine of the dimethylamino moiety of the macrolide's desosamine must be protonated under physiological pH, it acts as an H-bond donor, whereas HygA is an H-bond acceptor. The distance between the H-bonding N and C atoms is 3.2–3.5 Å, pointing to the rather weak character of this H-bond. Nevertheless, this additional interaction could play a significant role in the HygA-promoted binding of macro-lides to the Erm-modified ribosome, rationalizing the ability of the PTC-targeting drug HygA to partially restore the cidality of ketolides against Erm-expressing bacteria (Fig. 6). Importantly, the direct inter-action of HygA with macrolides on Erm-modified ribosome suggests that enhanced target engagement can lead to improved activity against multidrug-resistant pathogens. Thus, forming new interactions not necessarily with the target (70S ribosome) but between the drugs on the target may prove a general strategy in the design of expanded-spectrum antibiotics or their tandems capable of defeating Erm-methylase-mediated macrolide resistance.

## Discussion

HygA is a small-molecule antibiotic that was re-discovered in 2021 for its selective action against Gram-negative spirochete bacteria, effi-ciently killing the pathogens *B. burgdorferi*, *Treponema pallidum*, and *Treponema denticola*, the causative agents of Lyme disease, syphilis, and periodontal disease, respectively[49]. HygA is selectively taken up by spirochetes through a conserved nucleoside transporter, leaving the gut microbiome unaffected. Due to its promising selectivity, activity, oral bioavailability, and safety profile, HygA is a good candidate for clinical trials, and preclinical evaluation of this compound is currently ongoing. In this work, we demonstrated that HygA is also active against strains of Gram-positive *S. pneumoniae*. Since macrolides are also highly active against *S. pneumoniae*[26] and *B. burgdorferi* (MICs: 0.03–0.008 μg/ml)[50], we anticipate that HygA-macrolide pairs could potentially target the same microorganisms and their relatives. Our biochemical, microbiological, and structural data show that HygA improves binding of macrolides to their canonical site within the NPET of either WT or macrolide-resistant Erm-methylated 70S ribosomes.

This potentiating effect of HygA is likely due to: (1) direct interactions with the ribosome-bound macrolide and (2) inducing rotation of nucleotide A2062 of the 23S rRNA that, in turn, is required for efficient macrolide binding.

The cooperativity we observed between HygA and macrolides is reminiscent of previously described binding synergy between strep-togramin A and B antibiotics, which also target ribosomal PTC and NPET, respectively[30,44]. Since both types of streptogramins are usually co-produced by the same bacterial hosts belonging to the *Strepto-myces* genus, we can assume that these two drug classes co-evolved to efficiently inhibit the growth of competitor bacterial species. Indeed, the pair of dalfopristin (streptogramin A class) and quinupristin (streptogramin B class) demonstrates synergistic bactericidal activity against Gram-positive and Gram-negative bacteria and, due to their high efficacy, the mixture of these drugs is successfully used in the clinic (under the brand name Synercid)[51]. In contrast, the natural pro-ducer of HygA, *Streptomyces hygroscopicus*, does not seem to have any genes responsible for macrolide biosynthesis, suggesting that HygA and macrolides have not co-evolved in nature. Therefore, the anti-bacterial potency and efficacy of the reported here HygA-macrolide pairs most likely could be further improved by rational design and chemical derivatization of HygA, a macrolide, or both. Another approach to enhance target engagement and improve binding of two drugs with adjacent binding sites, such as HygA and macrolides, is to join them to each other covalently. Rib-X Pharmaceuticals (now Mel-linta Therapeutics) has previously utilized this strategy to synthesize RX-2102, a chimera of PTC-targeting florfenicol and the NPET-binding macrolide azithromycin, which exhibited excellent antibacterial activity against macrolide-resistant streptococci[52]. Therefore, it may be valuable to determine if the chemical fusion of HygA with a macrolide could yield a compound that is superior in its inhibitory and anti-bacterial properties compared to the parent drugs. We firmly believe that our work lays the foundation for the subsequent development of synergistic antibiotic tandems and drug chimeras with improved bactericidal properties against drug-resistant pathogens, such as those expressing *erm* genes. Furthermore, the conceptual strategy described here can be extended to other antibiotic sites on the 70S ribosome, such as the decoding center, as well as other essential drug targets within the bacterial cells.

## Methods

### Reagents

Non-radiolabeled erythromycin, azithromycin, chloramphenicol, clindamycin, and linezolid were purchased from MilliporeSigma (USA). Unless stated otherwise, all other reagents and chemicals were obtained from MilliporeSigma (USA). Hygromycin A was isolated and purified from *S. hygroscopicus* strain NRRL-2388 as described previously[49,53]. A201A was kindly provided by Dr. Daniel Wilson. Ketolides (telithromycin and solithromycin) were provided by Cempra Inc (USA). Radioactively labeled [14C]-ERY was obtained from American Radiolabeled Chemicals (USA).

### Competition binding assay

*S. pneumoniae* 70S ribosomes were prepared from Cp2000 strain as previously described[54]. Purified ribosomes were stored in a buffer containing 20 mM HEPES-KOH (pH 7.6), 50 mM $NH_4CH_3COO$, 6 mM $Mg(CH_3COO)_2$, and 4 mM β-mercaptoethanol. *S. pneumoniae* 70S ribosomes (3 nM) were mixed with 6 nM of [14C]-ERY in 1.5 ml of binding buffer (20 mM Tris-HCl (pH 7.6), 10 mM $MgCl_2$, 150 mM $NH_4Cl$, and 6 mM β-mercaptoethanol) and incubated for 30 min at 37 °C. Increasing concentrations of non-radioactive PTC inhibitors (chloramphenicol, hygromycin A, A201A, clindamycin, and linezolid) were added to the pre-formed 70S-ERY complexes and incubated at 37 °C for additional 2 h. To isolate ribosomes, 10 μl of 50 mg/ml suspension of DEAE magnetic beads (Bioclone, USA) was added to the reactions. After 15-min incubation at room temperature, the beads with immobilized ribosomes were captured using a magnetic stand (Invitrogen). The supernatant containing unbound [14C]-ERY was removed, and the beads were resuspended in 100 μl of 10 mM EDTA to release 70S ribosomes back in the solution. The [14C]-ERY-containing supernatant was then transferred to vials containing 5 ml of Ultima Gold liquid scintillation fluid (PerkinElmer), and the amount of remaining ribosome-associated radioactivity was quantified in a scintillation counter (Fig. 1e). Plotting and fitting of experimental data was performed using GraphPad Prism 9.3.1 software (GraphPad Software, Inc).

### Equilibrium binding of erythromycin to the 70S ribosome

*S. pneumoniae* 70S ribosomes were diluted to 5 nM, mixed with increasing concentrations of radiolabeled [14C]-ERY in the absence or presence of 100 μM of HygA in 1.5 ml of binding buffer (20 mM Tris-HCl (pH 7.6), 10 mM $MgCl_2$, 150 mM $NH_4Cl$, and 6 mM β-mercaptoethanol) and incubated at 37 °C for 2 h. 70S ribosomes were captured using DEAE magnetic beads (Bioclone, USA). The beads were rapidly washed three times with 0.75 ml of ice-cold binding buffer. The ribosomes were eluted using 10 mM EDTA and the amount of the remaining ribosome-associated radioactivity was quantified in a scintillation counter as described above. The resulting dissociation constants ($K_d$) were calculated using GraphPad Prism 9.3.1 software (Fig. 2a).

### Measurements of the rate of erythromycin dissociation from the 70S ribosome

The dissociation kinetics of [14C]-ERY from *S. pneumoniae* 70S ribosomes was measured in the absence or presence of 100 μM of HygA as previously described[26]. The resulting dissociation rate constants ($k_{off}$) were calculated using GraphPad Prism 9.3.1 software (Fig. 2b).

### Toe-printing analysis

The toe-printing analysis of drug-dependent ribosome stalling was carried out using synthetic *ermDL* DNA template as previously described[55–57] with minor modifications. Toe-printing reactions were carried out in 5-μl aliquots containing PURExpress transcription-translation coupled system (New England Biolabs, USA) to which the DNA template was added[58]. The reactions were incubated at 37 °C for 20–25 min. Reverse transcription on the templates was carried out using radioactively labeled primer NV1 (5′-GGTTATAATGAATTTT

GCTTATTAAC-3′). Primer extension products were resolved on 6% denaturing sequencing gels. The final concentrations of drugs were: 100 μM HygA and 50 μM ERY. In all reactions, we used 50 μM mupirocin (inhibitor of isoleucyl-tRNA-synthetase) to arrest ribosomes at the Ile codon downstream of the macrolide arrest site (Supplementary Fig. 1).

### MIC and checkerboard assays

Minimal inhibitory concentrations (MICs) of antibiotics against wild-type Cp2000 and *erm*-positive Cp1290 strains of *S. pneumoniae* were determined as described previously (Supplementary Table 2)[26]. In addition, MICs of antibiotics against *Borrelia burgdorferi* strain expressing GFP were measured using a plate reader after 1 week of incubation (Supplementary Table 2). Standard checkerboard assays were used to assess the interactions between two antibiotics in a two-dimensional array of 2-fold titrations of each drug in a given pair. The resulting fractional inhibitory concentration (FIC) indexes for a given HygA-macrolide combination acting against specific strains of *S. pneumonia* and *B. burgdorferi* were calculated as described previously[48]. First, the individual FICs were calculated by dividing the MIC of an antibiotic tested in combination by the MIC of the same antibiotic alone. Then, the FIC indexes were calculated by adding the FICs of each paired antibiotic at multiple concentrations (FIC index = FIC of antibiotic A + FIC of antibiotic B). The obtained FIC indexes of each HygA-macrolide pair were fitted into a non-linear regression curve using GraphPad Prism 9.3.1 software and subsequently interpreted based on synergy criteria to determine the degree of drug synergy/cooperativity[30,33,48,59–61]. A value of FIC index ≤0.5 reflects the definitive antimicrobial synergy between two drugs, whereas FIC index values between 0.5 and 1.0 point to strong additivity in their action. Conversely, FIC index values above 1.0 indicate antagonism in drug action.

### Time-kill assay

Assessment of bactericidal activity of antibiotics alone and in combinations was carried out according to the guidelines from the National Committee for Clinical Laboratory Standards[62]. The ability to kill macrolide-susceptible Cp2000 or macrolide-resistant *erm*-positive Cp1290 *S. pneumoniae* strains by antibiotics alone and in combinations was tested at different concentrations of macrolide and ketolide drugs in the presence of antibiotic concentrations at 4x the MIC of HygA (4 μg/ml for both *S. pneumoniae* strains). The data was plotted using GraphPad Prism 9.3.1 software (Figs. 5 and 6 and Supplementary Fig. 5).

### X-ray crystallographic structure determination

In this work, we used *T. thermophilus* 70S ribosomes complexed with *E. coli* protein Y merely as a tool to obtain structures of higher resolution. This approach is based on our previous finding that binding of PY to a vacant 70S ribosome stabilizes it by locking the head of the 30S subunit in an unrotated state, which leads to overall better diffraction and substantially higher structural resolution[39,40]. Additionally, by competing with the binding of mRNAs and tRNAs to the ribosome, PY purges any residual ribosome-bound tRNAs carried over during ribosome purification, thereby increasing the homogeneity of the ribosomes and, thus, improving their crystallizability and diffraction[42]. Since the binding site of PY is located on the small ribosomal subunit, it does not interfere with the binding of any ribosomal antibiotics targeting PTC or NPET located at the heart of the large ribosomal subunit. As a result, by using ribosome-PY complexes, we obtained significantly higher resolution and overall better quality electron density maps of the ribosome-bound HygA and macrolide drugs than possible otherwise.

Wild-type 70S ribosomes from *T. thermophilus* (strain HB8) containing unmodified residue A2058 of the 23S rRNA were prepared as described previously[40,42,63]. Purification of the Erm-modified 70S ribosomes from *T. thermophilus* (strain HB27 expressing Erm-like enzyme

from *Bifidobacterium thermophilum*) containing N6-dimethylated A2058 residue in the 23S rRNA was accomplished as optimized previously[7,20]. Ribosome complexes with *E. coli* protein Y, and HygA-macrolide pairs were formed by mixing 5 μM *Tth* 70S ribosomes with 50 μM PY, followed by the addition of HygA (350 μM) and either ERY (350 μM), AZI (700 μM), or TEL (350 μM). All *Tth* 70S ribosome complexes were formed in the buffer containing 5 mM HEPES-KOH (pH 7.6), 50 mM KCl, 10 mM $NH_4Cl$, and 10 mM $Mg(CH_3COO)_2$, and then crystallized in the buffer containing 100 mM Tris-HCl (pH 7.6), 2.9% (v/v) PEG-20K, 9–10% (v/v) MPD, 175 mM arginine, 0.5 mM β-mercaptoethanol. Crystals were grown by the vapor diffusion method in sitting drops at 19 °C, stabilized and cryo-protected stepwise using a series of buffers with increasing MPD concentrations (25, 30, and 35%) until reaching the final concentration of 40% (v/v) MPD as described previously[20,39,40,42,63,64]. The antibiotics were also added to the last 40% MPD stabilization buffers to 250 μM final concentration each. After stabilization, crystals were flash-frozen using a nitrogen cryo-stream at 80 °K (Oxford Cryosystems, UK).

Collection and processing of the X-ray diffraction data, model building, and structure refinement were performed as described in our previous publications[20,39,40,42,63,64]. The statistics of data collection and refinement are compiled in Supplementary Table 1. All figures showing atomic models were rendered using PyMOL Molecular Graphics System software (version 1.8.6, Schrödinger, www.pymol.org).

### Reporting summary

Further information on research design is available in the Nature Portfolio Reporting Summary linked to this article.

## Data availability

Coordinates and structure factors were deposited in the RCSB Protein Data Bank with accession codes: 8FC1 (wild-type *T. thermophilus* 70S ribosome in complex with protein Y, hygromycin A, and erythromycin), 8FC2 (wild-type *T. thermophilus* 70S ribosome in complex with protein Y, hygromycin A, and azithromycin), 8FC3 (wild-type *T. thermophilus* 70S ribosome in complex with protein Y, hygromycin A, and telithromycin), 8FC4 (A2058-N6-dimethylated *T. thermophilus* 70S ribosome in complex with protein Y, hygromycin A, and erythromycin), 8FC5 (A2058-N6-dimethylated *T. thermophilus* 70S ribosome in complex with protein Y, hygromycin A, and azithromycin) and 8FC6 (A2058-N6-dimethylated *T. thermophilus* 70S ribosome in complex with protein Y, hygromycin A, and telithromycin). All previously published structures that were used in this work for model building and structural comparisons were retrieved from the RCSB Protein Data Bank: PDB entries 6XHX, 5DOY, 4Z3S, 4V7V, 7S1G, and 4Y4O. Source data are provided with this paper.

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

## Acknowledgements

We thank Drs. Alexander Mankin and Nora Vazquez-Laslop for critical reading of the manuscript and valuable suggestions; Dr. Daniel Wilson for providing A201A antibiotic; the staff at NE-CAT beamlines 24ID-C and 24ID-E for help with data collection, especially Drs. Malcolm Capel, Frank Murphy, Surajit Banerjee, Igor Kourinov, David Neau, Jonathan Schuermann, Narayanasami Sukumar, Anthony Lynch, James Withrow, Kay Perry, Ali Kaya, and Cyndi Salbego. This work is based upon research conducted at the Northeastern Collaborative Access Team beamlines, which are funded by the National Institute of General Medical Sciences from the National Institutes of Health [P30-GM124165 to NE-CAT]. The Eiger 16M detector on 24-ID-E beamline is funded by an NIH-ORIP HEI grant [S10-OD021527 to NE-CAT]. This research used resources of the Advanced Photon Source, a US Department of Energy (DOE) Office of Science User Facility operated for the DOE Office of Science by Argonne National Laboratory under Contract No. DE-AC02-06CH11357. This work was supported by the National Institute of General Medical Sciences of the National Institutes of Health [R01-GM132302 to Y.S.P.], National Institute of Allergy and Infectious Diseases of the National Institutes of

Health [R01-AI162961 to Y.S.P.; R01-AI152210 to K.L.; R01-AI173064 to Z.P.B.; R21-AI137584 to Y.S.P.], the Illinois State startup funds [to Y.S.P.], and the Steven and Alexandra Cohen Foundation [to K.L.]. The funders had no role in study design, data collection and analysis, decision to publish or preparation of the manuscript.

## Author contributions

C.W.C. performed drug binding assays using *S. pneumoniae* and *T. thermophilus* 70S ribosomes; C.W.C. performed checkerboard assay on *S. pneumoniae* cells; N.L. carried out checkerboard experiments on *B. burgdorferi* cells; C.W.C. and Z.P.B. performed time-kill assay on *S. pneumoniae*; C.D. performed toe-printing assay; C.W.C., E.A.S., Y.S.P., and M.S.S. designed and performed X-ray crystallography experiments; K.L., M.S.S., and Y.S.P. supervised the experiments; all authors interpreted the results; C.W.C., Y.S.P., and M.S.S. wrote the manuscript.

## Competing interests

The authors declare no competing interests.
