## [Peer Review File · Nature Communications]

Structural insights into the mechanism of overcoming Erm-mediated resistance by macrolides acting in tandem with hygromycin AREVIEWER COMMENTS

Reviewer #1 (Remarks to the Author):

The prevalence of multidrug-resistant pathogens is a major public health threat that requires action on several levels, including the development of effective strategies to treat drug-resistant infections. One such strategy is to judiciously combine already approved pairs of antibiotics to make them more effective against resistant bacteria.

In this work, Chen et al. analyzed available high-resolution structures of ribosomes in complex with antibiotics targeting the peptidyl transferase center and nascent polypeptide exit tunnel to find pairs of antibiotics that might bind cooperatively to the ribosome, thereby increasing their potency. The authors convincingly show that desosamine-containing macrolides bind cooperatively with Hygromycin A (HygA) to the ribosome in vitro. This may be justified structurally by the ability of HygA to prime the 23S ribosomal RNA (rRNA) for macrolide binding, and through the formation of weak contacts between the two drugs. Moreover, this cooperativity is also observed on susceptible bacterial cells, where HygA is shown to increase the cidal activity of already bactericidal ketolides and convert bacteriostatic macrolides into bactericidal drugs. Remarkably, the authors then show, both biochemically and structurally, that HygA restores the ability of macrolides to bind to and inhibit ribosomes rendered drug-resistant through dimethylation of 23S rRNA residue A2058.

Overall this is a very elegant study that makes a beautiful use of high-resolution X-ray crystallography and biochemical approaches to reveal the mechanism by which HygA can potentiate the activity of macrolide antibiotics against resistant ribosomes. The results of this study should enable the rational design of HygA/macrolide pairs that are effective against multidrug resistant pathogens.

The manuscript is well and clearly written. Experiments are performed to high standards and the conclusions are fully justified by the data presented. As a result, I only have the following minor comments and suggestions that the authors may want to consider to improve the readability of their manuscript.

- Some additional background on HygA in the introduction/discussion would be helpful. For example, is HygA a clinically-useful antibiotic? Which organisms does it target and which organisms might be targeted by a HygA-macrolide pair?

- Companies such as Rib-X (now Mellinta Therapeutics) have focused on developing drugs that have increased binding to the ribosome by covalently joining two drugs with neighboring binding sites. The authors might consider adding a paragraph to the discussion to explore this possibility with HygA/macrolide pairs. In doing so, they may wish to consider the following points: (i) Based on the available structural data, is it likely that such a hybrid compound would easily find its way to its binding site? (ii) Given its physicochemical properties, would a hybrid compound of this kind be likely to find its way into bacterial cells? (iii) Which type of chemistry might be used to link the two drugs and how practical would this be from a chemical synthesis point of view?

- I would suggest making Figure S1A the first panel of Figure 1

- Line 153 – Briefly describe what the previously published technique is, possibly with the help of an explanatory diagram.

- How many times were the experiments in Figure 1A, B, etc repeated?

- Figure 1C – the vertical axis title is missing the “r” in ribosome

- Line 178 – The interaction between the drug and A2451 is mentioned but not shown in Fig. 3B.

- The panel order in Fig. 3 is confusing. I suggest that the authors swap the A and B labels around.

- Line 191 – The text should refer to Fig. 2D-F, not 3D-F.

- The authors may want to consider moving panels A, B and D of Fig. S5 to Fig. 4.

Reviewer #2 (Remarks to the Author):

The authors present here a coherent study of molecular mechanisms for synergistic actions of the two classes of ribosome targeting antibiotics, namely hygromycin A and macrolides, which target PTC and ribosomal exit tunnel, respectively. The study is very well executed and timely in the perspective of global antibiotic resistance scenario. The authors not only show with high resolution crystal structures how these drugs bind cooperatively to the bacterial ribosomes, but also demonstrate with biochemistry and in vivo assays their mode of action and bactericidal property. This study makes one hopeful that such combinatorial therapy could be of potential use for clinical cases. I have only few comments as written below.

1. It is unclear why the authors chose to use *E. coli* protein Y (PY) bound Tth 70S ribosomes for the crystal structures. Although the binding site of PY is in the small subunit it cannot be fully excluded that binding of EC PY may stabilize the drugs in their respective binding sites.
2. The use of EC PY in the crystal structures also compromises with the relevance in vivo, biochemistry and structural study correlations. At least one control competition binding assay should be performed (and added to Supplementary data) with Tth 70S ribosomes in the presence and absence of Ec PY.
3. The authors used the following concentrations of the drugs HygA (350 μ M) and either ERY (350 μ M), AZI (700 μ M), or TEL (350 μ M) in their crystallization reactions where they have only 5 μ M Tth 70S ribosomes. These drug concentrations are significantly higher than the corresponding MIC values. What is the rationale for using so high concentration of the drugs? Can the authors comment about any potential artifact caused by such a high (nonphysiological) concentrations?
4. From the Methods section for Time-kill assays it is very hard to see which concentration of drugs were enough for killing bacteria. It will be good to present those concentrations in microM in a table.
5. Since the PTC and NPET drugs inhibit protein synthesis, which mechanism likely dominates the protein synthesis inhibition scenario when added together?
6. The source of [¹⁴C]-ERY should be mentioned in 'reagents'
7. Figure 5: X-axis label missing.

Response to Reviewer #1

Remarks to the Authors:

The prevalence of multidrug-resistant pathogens is a major public health threat that requires action on several levels, including the development of effective strategies to treat drug-resistant infections. One such strategy is to judiciously combine already approved pairs of antibiotics to make them more effective against resistant bacteria.

In this work, Chen et al. analyzed available high-resolution structures of ribosomes in complex with antibiotics targeting the peptidyl transferase center and nascent polypeptide exit tunnel to find pairs of antibiotics that might bind cooperatively to the ribosome, thereby increasing their potency. The authors convincingly show that desosamine-containing macrolides bind cooperatively with Hygromycin A (HygA) to the ribosome in vitro. This may be justified structurally by the ability of HygA to prime the 23S ribosomal RNA (rRNA) for macrolide binding, and through the formation of weak contacts between the two drugs. Moreover, this cooperativity is also observed on susceptible bacterial cells, where HygA is shown to increase the cidal activity of already bactericidal ketolides and convert bacteriostatic macrolides into bactericidal drugs. Remarkably, the authors then show, both biochemically and structurally, that HygA restores the ability of macrolides to bind to and inhibit ribosomes rendered drug-resistant through dimethylation of 23S rRNA residue A2058.

Overall this is a very elegant study that makes a beautiful use of high-resolution X-ray crystallography and biochemical approaches to reveal the mechanism by which HygA can potentiate the activity of macrolide antibiotics against resistant ribosomes. The results of this study should enable the rational design of HygA/macrolide pairs that are effective against multidrug resistant pathogens. The manuscript is well and clearly written. Experiments are performed to high standards and the conclusions are fully justified by the data presented. As a result, I only have the following minor comments and suggestions that the authors may want to consider to improve the readability of their manuscript.

- 1. Some additional background on HygA in the introduction/discussion would be helpful. For example, is HygA a clinically-useful antibiotic? Which organisms does it target and which organisms might be targeted by a HygA-macrolide pair?*

Response: We are grateful to the reviewer for raising these great questions, and we completely agree that such information should have been included in the manuscript.

The small molecule hygromycin A was re-discovered in 2021 (Leimer et al., 2021), for its selective action against Spirochaete bacteria, efficiently killing *Borrelia burgdorferi*, *Treponema pallidum*, and *Treponema denticola*, the causative agents of Lyme disease, syphilis and periodontal disease, respectively. Hygromycin A targets the ribosome and is selectively taken up by Spirochetes through a conserved nucleoside transporter, leaving the gut microbiome largely unaffected. Selectivity, good efficacy, oral availability, and excellent safety make hygromycin A a good candidate for development and preclinical evaluation of this compound is currently ongoing. In this work, we demonstrated that HygA is also active against strains of Gram-positive *S. pneumoniae*. Macrolides, such as erythromycin, clarithromycin, or azithromycin, are highly active against *B. burgdorferi* with MICs of 0.03-0.008 µg/ml and were also shown to be bactericidal (Dever et al., 1992). Therefore, we anticipate that HygA-macrolide pairs could potentially target the same microorganisms.

Following the reviewer's suggestion, we have added this information at the beginning of the "Discussion" section.

2. *Companies such as Rib-X (now Mellinta Therapeutics) have focused on developing drugs that have increased binding to the ribosome by covalently joining two drugs with neighboring binding sites. The authors might consider adding a paragraph to the discussion to explore this possibility with HygA/macrolide pairs. In doing so, they may wish to consider the following points: (i) Based on the available structural data, is it likely that such a hybrid compound would easily find its way to its binding site? (ii) Given its physicochemical properties, would a hybrid compound of this kind be likely to find its way into bacterial cells? (iii) Which type of chemistry might be used to link the two drugs and how practical would this be from a chemical synthesis point of view?*

Response: We thank the reviewer for this comment – we agree that this discussion should have been included! Following this suggestion of the reviewer, we have added the following paragraph to the "Discussion" section of the manuscript:

"Another approach to enhance target engagement and improve binding of two drugs with adjacent binding sites, such as HygA and macrolides, is to join them to each other covalently. Rib-X Pharmaceuticals (now Mellinta Therapeutics) has previously utilized this strategy to synthesize RX-2102, a chimera of PTC-targeting florfenicol and NPET-binding macrolide azithromycin, which exhibited excellent antibacterial activity against macrolide-resistant streptococci. Therefore, it is very curious to check if the chemical fusion of HygA with a macrolide could yield a compound that is superior in its inhibitory and antibacterial properties compared to the parent drugs."

Because our primary expertise is outside the organic chemistry field, we prefer not to elaborate on the possible types of chemistry that might be used for linking HygA and a macrolide together. We hope that our message will be received by medicinal chemists eager to fuse these two drugs together. Also, we would like to avoid unnecessary speculations in the main text regarding whether or not a possible HygA-macrolide chimera could find its way into the cell and its binding site on the ribosome. We see no reasons why this won't be possible. Of course, the uptake of a chimeric molecule with an increased molecular weight could be an issue. However, this potential problem could be solved by finding the appropriate linker. Furthermore, given the results of our recent study showing the peptidyl-tRNAs carrying up to 7-aa-long peptides find their way into the ribosomal tunnel and adopt natural conformations (Syroegin *et al.*, 2022a), we expect that parts of hypothetical HygA-macrolide chimera would easily gain access to the corresponding sites in the PTC and NPET.

3. *I would suggest making Figure S1A the first panel of Figure 1.*

Response: Indeed, Figure S1A illustrates one of the key *in silico* findings that instigated the entire project. We agree with the reviewer that moving Figure S1A to the main text would definitely benefit the reader. However, we feel that this panel is not as informative as part of the entire figure S1A. Since the format of *Nature Communications* allows up to ten display items in the main text, we decided to move the Figure S1 as a whole to the main text, which now appears as Figure 1 in the revised version of the manuscript.

4. *Line 153: Briefly describe what the previously published technique is, possibly with the help of an explanatory diagram.*

Response: Following the reviewer's suggestion, we added two schematic diagrams explaining equilibrium binding and dissociation rate assays that we used to obtain data shown

in Figures 1B and C (now Figures 2A and B in the revised version). We have also added brief descriptions of the techniques to the figure legend.

5. *How many times were the experiments in Figure 1A, B, etc repeated?*

Response: We apologize for missing this important point and not making this clear. For each of the ribosome binding/dissociation experiments shown in Figure 1B-C (now Figure 2A-B in the revised version), three independent repeats were done. This information is now provided in the figure legend.

6. *Figure 1C: The vertical axis title is missing the “r” in ribosome.*

Response: Thank you for catching this glitch! The error is now corrected.

7. *Line 178: The interaction between the drug and A2451 is mentioned but not shown in Fig. 3B.*

Response: We agree with the reviewer that it would have been great to include nucleotide A2451 in Figure 3B (new Figure 4B). However, if we add nucleotide A2451 to that panel, it would appear in front of the viewer and block most of the HygA binding site view. Therefore, to preserve the clarity of the figure, we decided not to include this nucleotide in this panel. However, to address this important point of the reviewer, we have generated an additional panel to illustrate specifically how HygA intercalates in the A-site cleft formed by the nucleotides A2451 and C2452 on one side and U2506 on the other.

8. *The panel order in Fig. 3 is confusing. I suggest that the authors swap the A and B labels around.*

Response: While addressing the previous comment of this reviewer, we have added an additional panel to Figure 3 (now Figure 4 in the revised version) and also completely rearranged their layout. We hope, that the arrangement of panels in the new version of this figure is not confusing anymore.

9. *Line 191: The text should refer to Fig. 2D-F, not 3D-F.*

Response: The reviewer is absolutely correct that the text actually referred to Figure 2D-F and not Figure 3. Thank you for catching this, which is now corrected!

10. *The authors may want to consider moving panels A, B and D of Fig. S5 to Fig. 4.*

Response: Figure S5 (now Figure S4 in the revised version) shows negative-control experiments with tetracycline to emphasize that the observed effect is HygA-specific and that drugs other than HygA are unable to enhance bactericidal properties of macrolides. While certainly communicating important results, we feel that moving three panels from Figure S5 to the main-text Figure 4 would unnecessarily overload the latter and, most importantly, distract the reader from the main message of Figure 4, showing that HygA potentiates bactericidal properties of several tested macrolides.

Response to Reviewer #2

Remarks to the Authors:

The authors present here a coherent study of molecular mechanisms for synergistic actions of the two classes of ribosome targeting antibiotics, namely hygromycin A and macrolides, which target PTC and ribosomal exit tunnel, respectively. The study is very well executed and timely in the perspective of global antibiotic resistance scenario. The authors not only show with high resolution crystal structures how these drugs bind cooperatively to the bacterial ribosomes, but also demonstrate with biochemistry and in vivo assays their mode of action and bactericidal property. This study makes one hopeful that such combinatorial therapy could be of potential use for clinical cases. I have only few comments as written below.

- 1. It is unclear why the authors chose to use *E. coli* protein Y (PY) bound *Tth* 70S ribosomes for the crystal structures. Although the binding site of PY is in the small subunit it cannot be fully excluded that binding of *Ec* PY may stabilize the drugs in their respective binding sites.*

Response: In this work, we used *Tth* 70S ribosomes complexed with *E. coli* protein Y (PY) merely as a tool to obtain structures of higher resolution. This approach is based on our previous finding that binding of PY to a vacant 70S ribosome stabilizes it by locking the head of the 30S subunit in an unrotated state, which leads to overall better diffraction and substantially higher structural resolution (Polikanov et al., 2012; Polikanov et al., 2015a). Additionally, by competing with the binding of mRNAs and tRNAs to the ribosome, PY purges any residual ribosome-bound tRNAs that were carried over during ribosome purification, thereby increasing the homogeneity of the ribosomes and, thus, improving their crystallizability and diffraction. Since the binding site of PY is located on the small ribosomal subunit, it does not interfere with the binding of any ribosomal antibiotics targeting PTC or NPET located at the heart of the large ribosomal subunit. As a result, by using ribosome-PY complexes, we obtained significantly higher resolution and overall better quality electron density maps of the ribosome-bound HygA and macrolide drugs than possible otherwise.

We used this approach in a number of our recent studies, in which we provided compelling evidence that binding of PY to the 30S subunit does not have any effect on the 50S subunit (Chen et al., 2021; Svetlov et al., 2021; Syroegin et al., 2022b; Tereshchenkov et al., 2018). Most importantly, superpositioning of the structures of various PTC- and NPET-targeting antibiotics in the context of vacant(empty), PY-bound, or mRNA/tRNA-containing 70S ribosome complexes does not reveal any differences in the drug binding sites. More specifically, comparison of the previous structures of HygA and ERY bound to the 70S ribosomes containing tRNAs with the new structure of the same drugs simultaneously bound to the 70S ribosome containing PY reveals no substantial differences in the positions of the antibiotics (please refer to the new Supplementary Figure 1), suggesting that presence of PY exhibits no effect to the binding of either HygA or ERY to the ribosome.

Although a shorter version of the above explanation has been provided in the main text before, to further address this critical point of the reviewer, we have included additional clarifications to the “*Online Methods*” section of the manuscript as well.

2. *The use of Ec PY in the crystal structures also compromises with the relevance in vivo, biochemistry and structural study correlations. At least one control competition binding assay should be performed (and added to Supplementary data) with Tth 70S ribosomes in the presence and absence of Ec PY.*

Response: As explained in our response to the previous critical point of this reviewer, by using PY, we were not trying to recapitulate any physiologically relevant states of the ribosome and instead used this trick only as a tool to obtain higher-resolution structures. The structures of HygA and macrolides in the context of functionally-relevant ribosome complexes have been reported before. It is important for this study that their binding sites are identical to those discovered here. Since studying the effect of PY on the binding properties of HygA and macrolides has not been the focus of this study, we feel that providing the control experiments suggested by the reviewer is unnecessary and could also mislead the reader.

3. *The authors used the following concentrations of the drugs HygA (350 μ M) and either ERY (350 μ M), AZI (700 μ M), or TEL (350 μ M) in their crystallization reactions where they have only 5 μ M Tth 70S ribosomes. These drug concentrations are significantly higher than the corresponding MIC values. What is the rationale for using so high concentration of the drugs? Can the authors comment about any potential artifact caused by such a high (nonphysiological) concentrations?*

Response: The reviewer's concern about the high concentrations of drugs used in our crystallization reactions exceeding MIC values (and also Kd values), leading to non-specific binding and possible artifacts, is totally justified. Indeed, binding to multiple sites has been previously described for several ribosome-targeting antibiotics, including negamycin (Polikanov *et al.*, 2014) and tetracycline (Pioletti *et al.*, 2001). However, even at very high concentrations, HygA (Polikanov *et al.*, 2015b) and macrolides (Svetlov *et al.*, 2019; Svetlov *et al.*, 2021) bind exclusively to their canonical binding sites, in perfect agreement with previous structural, biochemical, and microbiological studies of these antibiotics.

Although unlikely to cause any structural artifacts, there are particular reasons for using such high concentrations in our structural studies. Firstly, the Kd values of HygA and macrolides/ketolides suggest that these drugs should bind completely to the ribosomes, even at 50-100-fold lower concentrations. However, the concentration of ribosomes increases significantly during crystal formation (from 5 μ M in the solution to 225 μ M in the crystal). Moreover, after crystals are formed, we stabilize and cryo-protect them before freezing, which includes four sequential steps of washing in solutions with increasing concentrations of precipitants. Therefore, we use high drug concentrations commensurate with the actual ribosome concentration in a crystal to sustain efficient binding during crystal growth and prevent unwanted dissociation during subsequent stabilization and cryoprotection steps.

Another reason for using high drug concentrations is related to our work on Erm-modified drug-resistant ribosomes containing A2058-N6-dimethylated residue. The affinity of macrolides to such ribosomes is unknown but is expected to be very low. Indeed, as evident from Figure 6A (now Figure 7A), almost no electron density can be observed for the ribosome-bound macrolide erythromycin, yet again suggesting that high concentrations of macrolides used in our structural experiments do not result, at least, in structural artifacts. The significantly better electron density for the ribosome-bound erythromycin observed in the presence of HygA on the same Erm-modified ribosome even further signifies that macrolide binding is extremely specific to their canonical site in the ribosomal tunnel. Nevertheless, based on our prior

experience, we could have actually used lower drug concentrations for the structural studies involving WT 70S ribosomes. Still, we used the same concentrations throughout to ensure that our structures of WT vs. Erm-modified ribosomes were consistent.

4. *From the Methods section for time-kill assays it is very hard to see which concentration of drugs were enough for killing bacteria. It will be good to present those concentrations in microM in a table.*

Response: We agree entirely with the reviewer and apologize for not including this important information before. As suggested by the reviewer, we have provided all MIC values for the drugs used in the checkerboard and time-kill assays in the new Supplementary Table 2 and also indicated the actually used concentrations used for the time-kill assay in the legends for Figures 4 and 5 (new Figures 5 and 6).

5. *Since the PTC and NPET drugs inhibit protein synthesis, which mechanism likely dominates the protein synthesis inhibition scenario when added together?*

Response: Excellent point – thank you! It has been shown previously that macrolides are context-specific translation inhibitors, meaning that instead of being indiscriminate inhibitors of every peptide bond formation, macrolides cause ribosome stalling only at particular sequence motifs within open reading frames (Kannan *et al.*, 2014). In contrast to macrolides, HygA appears to be an indiscriminate inhibitor of the first peptide bond formation on various mRNA templates (Leimer *et al.*, 2021; Polikanov *et al.*, 2015b). Therefore, when added together, we expect HygA to dominate and inhibit translation at the start codon way before ribosome reaches the macrolide-arrest sequence further in the mRNA's ORF. Indeed, this prediction is supported by the *in vitro* toe-printing assay, showing that, in the presence of HygA, ribosomes predominantly become arrested at the start codon and do not progress into the ORF (see the new Supplementary Fig. 1 in the revised version of the manuscript). To further address this critical comment of the reviewer we have added a few sentences to the results section referencing these new data.

Here we would like to point out that the effect of ribosome-targeting antibiotics on bacterial cell growth and viability could be defined by the kinetics of drug interaction with the ribosome rather than its ability to inhibit protein synthesis in the cell globally. Previously, we have illustrated this principle for ketolides, which are less potent inhibitors of global cellular translation compared to macrolide erythromycin, nevertheless possess superior antibacterial activity due to their extremely slow dissociation from the ribosome and, as a result, prolonged translation shutdown leading to cell death (Svetlov *et al.*, 2020; Svetlov *et al.*, 2017). Thus, the observed here 60-fold slower dissociation rate of macrolides from the ribosome in the presence of HygA is likely to be the cause of the improved antibacterial potency of the HygA/macrolide drug pairs rather than their ability to better inhibit protein synthesis. We have discussed these critical points in the “*Results*” section of the manuscript.

6. *The source of [¹⁴C]-ERY should be mentioned in ‘reagents’.*

Response: The radioactively labeled [¹⁴C]-ERY was obtained from *American Radiolabeled Chemicals (ARC)*. This information has been included in the “*Reagents*” section of the *Methods*.

7. *Figure 5: X-axis label missing.*

Response: Thank you for catching this glitch! We somehow cropped the image above the axis title, which is now added to the figure.

REFERENCES:

Chen, C.W., Pavlova, J.A., Lukianov, D.A., Tereshchenkov, A.G., Makarov, G.I., Khairullina, Z.Z., Tashlitsky, V.N., Paleskava, A., Konevega, A.L., Bogdanov, A.A., *et al.* (2021). Binding and action of triphenylphosphonium analog of chloramphenicol upon the bacterial ribosome. *Antibiotics (Basel)* *10*.

Dever, L.L., Jorgensen, J.H., and Barbour, A.G. (1992). *In vitro* antimicrobial susceptibility testing of *Borrelia burgdorferi*: a microdilution MIC method and time-kill studies. *J. Clin. Microbiol.* *30*, 2692-2697.

Kannan, K., Kanabar, P., Schryer, D., Florin, T., Oh, E., Bahroos, N., Tenson, T., Weissman, J.S., and Mankin, A.S. (2014). The general mode of translation inhibition by macrolide antibiotics. *Proc. Natl. Acad. Sci. USA* *111*, 15958-15963.

Leimer, N., Wu, X., Imai, Y., Morrisette, M., Pitt, N., Favre-Godal, Q., Iinishi, A., Jain, S., Caboni, M., Leus, I.V., *et al.* (2021). A selective antibiotic for Lyme disease. *Cell* *184*, 5405-5418.

Pioletti, M., Schlünzen, F., Harms, J., Zarivach, R., Glühmann, M., Avila, H., Bashan, A., Bartels, H., Auerbach, T., Jacobi, C., *et al.* (2001). Crystal structures of complexes of the small ribosomal subunit with tetracycline, edeine and IF3. *EMBO J* *20*, 1829-1839.

Polikanov, Y.S., Blaha, G.M., and Steitz, T.A. (2012). How hibernation factors RMF, HPF, and YfiA turn off protein synthesis. *Science* *336*, 915-918.

Polikanov, Y.S., Melnikov, S.V., Soll, D., and Steitz, T.A. (2015a). Structural insights into the role of rRNA modifications in protein synthesis and ribosome assembly. *Nat. Struct. Mol. Biol.* *22*, 342-344.

Polikanov, Y.S., Starosta, A.L., Juette, M.F., Altman, R.B., Terry, D.S., Lu, W., Burnett, B.J., Dinos, G., Reynolds, K.A., Blanchard, S.C., *et al.* (2015b). Distinct tRNA accommodation intermediates observed on the ribosome with the antibiotics hygromycin A and A201A. *Mol Cell* *58*, 832-844.

Polikanov, Y.S., Szal, T., Jiang, F., Gupta, P., Matsuda, R., Shiozuka, M., Steitz, T.A., Vázquez-Laslop, N., and Mankin, A.S. (2014). Negamycin interferes with decoding and translocation by simultaneous interaction with rRNA and tRNA. *Mol Cell* *56*, 541-550.

Svetlov, M.S., Cohen, S., Alsuhebany, N., Vazquez-Laslop, N., and Mankin, A.S. (2020). A long-distance rRNA base pair impacts the ability of macrolide antibiotics to kill bacteria. *Proc. Natl. Acad. Sci. USA* *117*, 1971-1975.

Svetlov, M.S., Plessa, E., Chen, C.W., Bougas, A., Krokidis, M.G., Dinos, G.P., and Polikanov, Y.S. (2019). High-resolution crystal structures of ribosome-bound chloramphenicol and erythromycin provide the ultimate basis for their competition. *RNA* *25*, 600-606.

Svetlov, M.S., Syroegin, E.A., Aleksandrova, E.V., Atkinson, G.C., Gregory, S.T., Mankin, A.S., and Polikanov, Y.S. (2021). Structure of Erm-modified 70S ribosome reveals the mechanism of macrolide resistance. *Nat. Chem. Biol.* *17*, 412-420.

Svetlov, M.S., Vazquez-Laslop, N., and Mankin, A.S. (2017). Kinetics of drug-ribosome interactions defines the efficacy of macrolide antibiotics. *Proc. Natl. Acad. Sci. USA* *114*, 13673-13678.

Syroegin, E.A., Aleksandrova, E.V., and Polikanov, Y.S. (2022a). Insights into the ribosome function from the structures of non-arrested ribosome-nascent chain complexes. *Nat. Chem.* *15*, 143-153.

Syroegin, E.A., Flemmich, L., Klepacki, D., Vazquez-Laslop, N., Micura, R., and Polikanov, Y.S. (2022b). Structural basis for the context-specific action of the classic peptidyl transferase inhibitor chloramphenicol. *Nat. Struct. Mol. Biol.* *29*, 152-161.

Tereshchenkov, A.G., Dobosz-Bartoszek, M., Osterman, I.A., Marks, J., Sergeeva, V.A., Kasatsky, P., Komarova, E.S., Stavrianidi, A.N., Rodin, I.A., Konevega, A.L., *et al.* (2018). Binding and action of amino acid analogs of chloramphenicol upon the bacterial ribosome. *J Mol Biol* *430*, 842-852.

REVIEWERS' COMMENTS

Reviewer #2 (Remarks to the Author):

The authors have clarified all my concerns in the revised version of the manuscript. It is indeed a strong study in the context of antibiotic resistance and should encourage others to use combination of antibiotics to overcome the problem.

Response to Reviewer #2

Remarks to the Authors:

The authors have clarified all my concerns in the revised version of the manuscript. It is indeed a strong study in the context of antibiotic resistance and should encourage others to use combination of antibiotics to overcome the problem.

Response: No response is needed.